# Deterministic Transport-Based Sampling via Wasserstein Gradient Flows

## Abstract

We study the problem of transport-based sampling for target distributions that are implicitly defined as minimizers of objective functions. This formulation generalizes existing approaches that rely on learning time-varying scores under specific divergences, such as KL minimization. Recent advances in score-based transport methods highlight several advantages, including smooth deterministic trajectories and monotone, noise-free convergence compared to Langevin dynamics. Motivated by these benefits, we develop a stochastic Wasserstein gradient flow framework, in which particle-based estimators approximate the Wasserstein gradient and transport an arbitrary initial distribution toward the target. We establish convergence analysis that account for the mean and variance of these stochastic gradient estimates. We further demonstrate applications to multi-objective optimization and particle transport, leveraging maximum mean discrepancy and Wasserstein distance as guiding metrics.

## 1 Introduction

Distribution transport studies how probability measures evolve smoothly from a source distribution $\mu_0$ to a target $\mu_1$ under geometric or variational principles Ilin et al. (2025); Lu et al. (2024); Boffi & Vanden-Eijnden (2023). A dynamical formulation represents transport as a path $(\mu_t)_{t \in [0,1]}$ governed by the continuity equation:

$$\partial_t \mu_t + \nabla \cdot (\mu_t v_t) = 0,$$

where $v_t$ is the velocity field driving the flow. This perspective connects naturally to *generative modeling* (Song & Ermon, 2019; Ho et al., 2020; Song et al., 2020), where the goal is to transform a simple prior (e.g., Gaussian noise) into realistic samples. Flow-based and diffusion models can be interpreted as constructing such transport dynamics, providing both theoretical foundations and computational tools for modern generative methods.

We focus on *deterministic transport-based sampling* studied by Ilin et al. (2025):

$$\frac{\mathrm{d}}{\mathrm{d}t} X_t = v_t(X_t), \quad X_t \sim \mu_t,$$

where the velocity field $v_t$ is the Wasserstein gradient of an objective function $\mathcal{L}(\mu)$. This framework generalizes score-based transport, which arises as the gradient flow of the Kullback-Leibler (KL) divergence, i.e., $\mathcal{L}(\mu) = \mathrm{KL}(\mu \,\|\, \nu)$, in the Wasserstein space for transporting a source distribution toward $\nu$. Our formulation offers two main advantages:

1. **Flexible geometry.** By selecting different objective functions, one can design transport trajectories that encode richer geometric structures beyond KL divergence minimization. In particular, we introduce Integral Probability Metric (IPM) regularization to mitigate out-of-distribution issues when score estimation degenerates near the source distribution (Section 3).

2. **Implicit target distribution.** Existing sampling approaches typically assume that the target distribution is explicitly specified and or samples from it are available. In contrast, our framework allows the target distribution to be defined implicitly as the minimizer of a chosen objective. We illustrate this with applications to sampling from Pareto fronts in multi-objective optimization (Section 4).

In Section 2, we establish convergence guarantees based on the mean and variance of the Wasserstein gradient, providing conditions under which the flow recovers the desired distribution.

**Motivations**   Denoising diffusion models (Song & Ermon, 2019; Ho et al., 2020; Song et al., 2020) are tailored to transport Gaussian priors into target distributions, but are not naturally suited for settings with non-Gaussian source distributions. While conditional flow matching (CFM) Tong et al. (2023); Pooladian et al. (2023) enables transport from arbitrary sources, its reliance on simple interpolation paths limits its ability to capture richer geometric structures. Moreover, neither framework accommodates cases where the target distribution is defined only implicitly as the minimizer of an objective.

Our proposed framework does not aim to outperform diffusion or CFM methods (which have demonstrated strong empirical performance), but instead to broaden the scope of transport-based sampling to more general objectives and implicit targets.

**Notations**   $\|\cdot\|_2$ denotes the Euclidean norm. Let $\mathcal{P}_2(\mathbf{R}^d)$ be the set of probability distributions with finite second moment on $\mathbf{R}^d$ and $\mathcal{P}_2^r(\mathbf{R}^d)$ be its subset of absolutely continuous distributions with respect to Lebesgue measure. For $\mu \in \mathcal{P}_2(\mathbf{R}^d)$, $L_2(\mu; \mathbf{R}^d)$ denotes the space of the $\mathbf{R}^d$-valued square integrable functions $\varphi$ with respect to $\mu$; $\int \|\varphi(x)\|_2^2 \mu(\mathrm{d}x) < \infty$, and $\langle \cdot, \cdot \rangle_{L_2(\mu;\mathbf{R}^d)}$ denotes the inner-product of this space; $\langle \varphi, \psi \rangle_{L_2(\mu;\mathbf{R}^d)} = \int \varphi(x)^\top \psi(x) \mu(\mathrm{d}x)$ for $\varphi, \psi \in L_2(\mu; \mathbf{R}^d)$. Ent denotes the negative entropy: $\mathrm{Ent}(\mu) = \int \mu(\mathrm{d}x) \log \frac{\mathrm{d}\mu}{\mathrm{d}x}(x)$.

## 2   WASSERSTEIN GRADIENT DESCENT

We present Wasserstein gradient descent over probability spaces and analyze its convergence under biased gradient estimates.

Let $\mathcal{P}_2^r(\mathbb{R}^d)$ denote the space of probability measures on $\mathbb{R}^d$ that are absolutely continuous with respect to Lebesgue measure and have a finite second moment. Given a functional $\mathcal{F} : \mathcal{P}_2^r(\mathbb{R}^d) \mapsto \mathbb{R}$, *Wasserstein gradient* $\nabla_W \mathcal{F}(\mu)(\cdot) : \mathbb{R}^d \to \mathbb{R}^d$ at the distribution $\mu_t \in \mathcal{P}_2^r(\mathbb{R}^d)$ indicates the direction in which the particle $X_t \sim \mu_t$ evolves to reduce $F(\mu_t)$, leading to the Wasserstein Gradient Flow (WGF):

$$\mathrm{d}X_t = -\nabla_W \mathcal{F}(\mu_t)(X_t)\mathrm{d}t, \ X_t \sim \mu_t. \tag{1}$$

The convergence of WGF can be then studied through an associated Fokker-Planck equation (Jordan et al., 1998), known as the continuity equation, which defines the evolution of $\mu_t = \mathrm{Law}(X_t)$ in $\mathcal{P}_2^r(\mathbb{R}^d)$:

$$\partial_t \mu_t + \nabla \cdot (\mu_t \nabla_W \mathcal{F}(\mu_t)) = 0. \tag{2}$$

**Stochastic Particle Gradient Descent**   We say the functional $\varphi : \mathcal{P}_2(\mathbb{R}^d) \to \mathbb{R}$ is Fréchet differentiable at $\mu \in \mathcal{P}_2^r(\mathbb{R}^d)$ regarding $L_2(\mu; \mathbb{R}^d)$ when there exists $\nabla_W \varphi(\mu) \in L_2(\mu; \mathbb{R}^d)$ such that for all $\nu \in \mathcal{P}_2(\mathbb{R}^d)$

$$\varphi(\nu) - \varphi(\mu) = \left\langle \nabla_W \varphi(\mu), t_\mu^\nu - id \right\rangle_{L_2(\mu;\mathbb{R}^d)} + o(W_2(\mu, \nu)),$$

where $t_\mu^\nu$ is the optimal transport: $\mu \mapsto \nu$. We refer to $\nabla_W \varphi(\mu)$ as the Wasserstein gradient of $\mathcal{F}$ at $\mu \in \mathcal{P}_2^r(\mathbb{R}^d)$.

We consider the following distribution optimization problems for minimizing a functional $\mathcal{L} : \mathcal{P}_2(\mathbb{R}^d) \to \mathbb{R}$:

$$\min_{\mu \in \mathcal{P}_2(\mathbb{R}^d)} \mathcal{L}(\mu). \tag{3}$$

We denote by $\mu_* \in \mathcal{P}_2(\mathbb{R}^d)$ the optimal solution of this problem.

Stochastic Particle Gradient Descent (SPGD) (Nitanda & Suzuki, 2017) is an iterative optimization method for solving this problem (see Appendix D for details). Let $\mu_0$ be an initial probability distribution. Then, SPGD implicitly updates the distribution through the particle update. Given a current distribution $X_k \sim \mu_k$, an iteration of SPGD is defined by

$$X_{k+1} = X_k - \eta G_k(X_k), \tag{4}$$

where $G_k(\cdot) : \mathbf{R}^d \to \mathbf{R}^d$ is a biased estimator to the Wasserstein gradient $\nabla_W \mathcal{L}(\mu_k)(\cdot)$. Therefore, SPGD can be seen as a stochastic optimization method that discretizes WGF in $\mathcal{P}_2(\mathbf{R}^d)$. The particle update (4) induces the evolution of $\mu_{k+1} = \text{Law}(X_k)$ as follows via transport map:

$$\mu_{k+1} = (id - \eta G_k)_\sharp \mu_k. \tag{5}$$

In practice, we approximate the update (4) using the finite-particle system $\{x_k^i\}_{i=1}^N \subset \mathbf{R}^d$ that constitutes an empirical distribution $\frac{1}{N}\sum_{i=1}^N \delta_{x_k^i} \sim \mu_k$, and each particle is updated as follows: for $i \in \{1, 2, \dots, N\}$,

$$x_{k+1}^i = x_k^i - \eta G_k(x_k^i). \tag{6}$$

Eq. (4) and (5) are then attained by taking the mean-field limit: $N \to \infty$. In this paper, we study the convergence capability of SPGD (4) in the infinite particle setting.

In the following, we present the convergence results of SPGD under geodesic convexity and the PL condition. We make the following generic assumption.

**Assumption 1.**

1. *SPGD performs in $\mathcal{P}_2^r(\mathbf{R}^d)$, that is, $\{\mu_k\}_{k=0}^\infty \subset \mathcal{P}_2^r(\mathbf{R}^d)$.*

2. *There exists $L \geq 0$ such that for all $\xi \in L_2(\mu; \mathbf{R}^d)$ and $\mu \in \mathcal{P}_2^r(\mathbf{R}^d)$, the following holds:*

$$\mathcal{L}((id + \xi)_\sharp \mu) \leq \mathcal{L}(\mu) + \langle \nabla_W \mathcal{L}(\mu), \xi \rangle_{L_2(\mu; \mathbf{R}^d)} + \frac{L}{2}\|\xi\|_{L_2(\mu; \mathbf{R}^d)}^2.$$

3. *There exists $\tau, \ \sigma^2 \geq 0$ such that for all $\mu \in \mathcal{P}_2^r(\mathbf{R}^d)$,*

$$\|\nabla_W \mathcal{L}(\mu_k) - \mathbf{E}[G_k]\|_{L_2(\mu_k; \mathbf{R}^d)}^2 \leq \tau,$$
$$\mathbf{E}_{G_k}[\|G_k - \mathbf{E}[G_k]\|_{L_2(\mu_k; \mathbf{R}^d)}^2] \leq \sigma^2.$$

**Remark 1.** *The second assumption is a counterpart of Lipschitz smoothness which is often supposed in the finite-dimensional optimization setting. The third assumption allows for the bias estimation of Wasserstein gradient up to a tolerance $\tau$ with a variance $\sigma^2$.*

**Convergence under geodesic convexity**   We assume $\mathcal{L}(\mu)$ satisfies Assumption 2.

**Assumption 2** (Geodesic convexity (Ambrosio et al., 2008))**.** *Given $c \geq 0$, we suppose $\mathcal{L}$ is $c$-geodesically convex. That is, for every pair $\mu_1, \mu_2 \in \mathcal{P}_2(\mathbf{R}^d)$, there exists an optimal coupling $\gamma$ between $\mu_1$ and $\mu_2$ such that*

$$\mathcal{L}(\mu_t) \leq (1-t)\mathcal{L}(\mu_1) + t\mathcal{L}(\mu_2) - \frac{c}{2}t(1-t)W_2^2(\mu_1, \mu_2),$$

*where $\mu_t = ((1-t)\pi_1 + t\pi_2)_\sharp \gamma$, $\pi_1$ and $\pi_2$ being the projections onto the first and second coordinate in $\mathbf{R}^d \times \mathbf{R}^d$, respectively.*

**Example 1.** *Examples include potential energy functional $\int f(x)\,d\mu(x)$ and interaction energies $\int\int w(x-y)\,d\mu(x)\,d\mu(y)$ for convex functions $f$ and $w$, internal energy functionals $\int \mu(x)^{m-1}\,d\mu(x)$ for $m > 1$ (Ambrosio et al., 2008).*

We here present the first convergence result of SPGD under geodesic convexity.

**Theorem 1.** *Under Assumptions 1 and 2, we run SPGD (4) with $\eta \leq \frac{1}{4L}$. If $c = \tau = 0$, then we get*

$$\frac{1}{T}\sum_{k=0}^{T-1}(\mathbf{E}[\mathcal{L}(\mu_k)] - \mathcal{L}(\mu_*)) \leq \frac{1}{2T}\left(\frac{1}{\eta}W_2^2(\mu_0, \mu_*) + 8\mathcal{L}(\mu_0)\right) + \eta\sigma^2.$$

*Moreover, if $c > 0$, we get*

$$\mathbf{E}[W_2^2(\mu_T, \mu_*)] + 8\eta(\mathbf{E}[\mathcal{L}(\mu_T)] - \mathcal{L}(\mu_*))$$
$$\leq \frac{4\tau}{c^2} + \frac{16\eta\tau}{c} + \frac{4\eta\sigma^2}{c} + \left(1 - \frac{c\eta}{2}\right)^T \left(W_2^2(\mu_0, \mu_*) + 8\eta(\mathcal{L}(\mu_0) - \mathcal{L}(\mu_*))\right).$$

The proof is provided in Appendix A. This theorem means that SPGD can converge up to $\eta\sigma^2$ error in case of $c = \tau = 0$ and up to $\frac{4\tau}{c^2} + \frac{16\eta\tau}{c} + \frac{4\eta\sigma^2}{c}$ error in generic case of $c > 0$ with the sublinear and linear convergence rates, respectively. Therefore, by appropriately controlling $\eta, \sigma^2, \tau$, it achieves a desired error $\epsilon$. In that way, we can estimate the iteration complexities to achieve $\epsilon$-accurate solution as follows: $O\left(\max\left\{\frac{L}{\epsilon}, \frac{\sigma^2}{\epsilon^2}\right\}\right)$ in the case of $c = \tau = 0$ and $O\left(\max\left\{L, \frac{\sigma^2}{c\epsilon}\log\frac{1}{\epsilon}\right\}\right)$ in the case of $c > 0$ under $\tau = c^2\epsilon$.

**Convergence under Polyak-Łojasiewicz condition**  We assume $\mathcal{L}(\mu)$ satisfies Assumption 3.

**Assumption 3** (Polyak-Łojasiewicz condition). *Given $c > 0$, we suppose $\mathcal{L}$ satisfies $c$-PL condition. That is, for all $\mu \in \mathcal{P}_2^r(\mathbf{R}^d)$, it follows that*

$$c(\mathcal{L}(\mu) - \mathcal{L}(\mu_*)) \leq \|\nabla_W \mathcal{L}(\mu)\|_{L_2(\mu;\mathbf{R}^d)}^2.$$

This condition implies that the Wasserstein gradient does not vanish unless the current distribution is optimal. Hence, we can expect that each iteration of SPGD, which approximates WGF, reduces the objective value by the gradient norm, ignoring some errors.

**Example 2.** *A notable examples is a linearly convex functional $F : \mathcal{P}_2(\mathbf{R}^d) \to \mathbf{R}$ with entropy regularization (see Appendix B for proof):*

$$\mathcal{L}(\mu) = F(\mu) + \lambda\mathrm{Ent}(\mu). \tag{7}$$

While it is noted that Yang et al. (2020) studied convergence under the PL condition, our result in Theorem 2 establishes convergence under biased Wasserstein gradient estimation in more general settings.

**Theorem 2.** *Under Assumption 1 and 3, we run SPGD (4) with $\eta \leq \frac{1}{4L}$. Then, we get*

$$\mathbf{E}[\mathcal{L}(\mu_T)] - \mathcal{L}(\mu_*) \leq \left(1 - \frac{\eta c}{4}\right)^T (\mathcal{L}(\mu_0) - \mathcal{L}(\mu_*)) + \frac{3\tau}{c} + \frac{2L\eta\sigma^2}{c}.$$

The proof is provided in Appendix C. From Theorem 2, we can deduce that the iteration complexity of SPGD to achieve $\epsilon$-accurate solution is $O\left(\max\left\{\frac{L\sigma^2}{c^2\epsilon}, \frac{L}{c}\right\}\log\frac{1}{\epsilon}\right)$, provide $\tau = c\epsilon$. In case of Example 2 with $\sigma = 0$, this iteration complexity is much smaller than the complexity $O\left(\frac{1}{\alpha^2\lambda^2\epsilon}\log\frac{1}{\epsilon}\right)$ of MFLD (Nitanda et al., 2022).

## 3 APPLICATION 1: FLOW WITH IPM REGULARIZATION

Deterministic distribution transport and sampling can be formulated as a Wasserstein gradient flow of the KL divergence $\mathrm{KL}(\mu \| \nu)$ (Boffi & Vanden-Eijnden, 2023; Ilin et al., 2025), which transports an initial distribution $\mu_0$ toward a target distribution $\nu$:

$$\frac{\mathrm{d}}{\mathrm{d}t}X_t = \nabla\log\mu_t(X_t) - \nabla\log\nu(X_t), \; X_t \sim \mu_t.$$

In principle, this flow converges to the target distribution $\nu$. However, directly simulating the dynamics is often unstable in practice. Prior works mitigate this by introducing *annealing paths* between source distribution $\mu_0$ and $\nu$, thereby smoothing the probability trajectory (Neal, 2001; Chehab et al., 2024; Guo et al., 2024). These approaches, however, primarily impose smoothness in the *information-geometric* sense and do not account for alternative structures, such as the *Euclidean geometry* of the sample space.

**A key challenge** arises when the supports of $\mu_0$ and $\nu$ are far apart. In such cases, score estimates of $\nu$ provide little guidance for particles initialized under $\mu_0$. Consequently, particle movement of the flow in the early stages is dominated by noise, leading to unstable or non-convergent trajectories.

To address this limitation, we propose augmenting the objective with a linearly convex probablity distance function, which aims to enrich the geometry of the Wasserstein gradient flow:

$$\mathcal{L}(\mu) = \mathcal{G}(\mu, \nu) + \lambda\mathrm{KL}(\mu \| \nu), \tag{8}$$

where $\mathcal{G}(\mu, \nu)$ is Integral Probability Metrics (IPM).

**Integral Probability Metrics (IPM)** Given two probability distributions $\mu$ and $\nu$ on $\mathcal{X}$, and a class of witness functions $\mathcal{F} : \mathcal{X} \to \mathbb{R}$, the IPM is defined as

$$\text{IPM}_{\mathcal{F}}(\mu, \nu) := \sup_{f \in \mathcal{F}} \left| \int_{\mathcal{X}} f(x) \, \mathrm{d}\mu(x) - \int_{\mathcal{X}} f(x) \, \mathrm{d}\nu(x) \right|. \tag{9}$$

The functional $\mathcal{L}(\mu) = \text{IPM}_{\mathcal{F}}(\mu, \nu)$ is *linearly convex* in the probability space $\mu \in \mathcal{P}_2^r(\mathcal{X})$ (see proof in Appendix E). Consequently, the combined objective (8) satisfies the PL condition as illustrated in Example 2. The discriminative power of an IPM depends on the choice of witness function class $\mathcal{F}$.

**Example 3.** *Total variation distance corresponds to $\mathcal{F} = \{f : \|f\|_{\infty} \leq 1\}$. Wasserstein-1 distance corresponds to $\mathcal{F} = \{f : \|f\|_{\text{Lip}} \leq 1\}$, where $\|f\|_{\text{Lip}}$ denotes the Lipschitz constant of $f$ (by Kantorovich-Rubinstein duality). Maximum Mean Discrepancy (MMD) corresponds to $\mathcal{F} = \{f : \|f\|_{\mathcal{H}} \leq 1\}$, where $\mathcal{H}$ is a reproducing kernel Hilbert space (RKHS).*

By selecting different $\mathcal{F}$, the flow can be regularized under geometries beyond information geometry. For example, MMD is known for its moment-matching properties, whereas the 1-Wasserstein distance measures the minimal work in the sample space required to transport the mass between distributions.

**Proposition 1** (Wasserstein gradient of IPMs). *Let $\mathcal{G}(\mu_t) = \text{IPM}_{\mathcal{F}}(\mu_t, \nu)$. Its Wasserstein gradient is given by*

$$\nabla \frac{\delta \mathcal{G}}{\delta \mu_t}(x) = \nabla f_t^*(x), \tag{10}$$

*where $f_t^*$ is the critic function that achieves the supremum in $\text{IPM}_{\mathcal{F}}(\mu_t, \nu)$.*

The proof is provided in Appendix F.

**Sup-MMD.** In general, the Wasserstein gradient does not admit a closed-form expression, as it requires solving for the critic function of the IPM. For kernel MMD (Galashov et al., 2024), however, the gradient can be written explicitly as $\mathbb{E}_{z \sim \mu_t}[\nabla_x k(x, z)] - \mathbb{E}_{y \sim \nu}[\nabla_x k(x, y)]$, where $k$ is the reproducing kernel. In practice, this can be efficiently approximated using sample averages. A limitation of standard MMD is that a fixed kernel often lacks sufficient discriminative power. To address this, we enrich the witness function class $\mathcal{F}$ by introducing a family of kernels, which we term Sup-MMD. Using Random Fourier Features (RFF), invariant kernels can be expressed as expectations of cosine features over frequency distributions, enabling maximization of the MMD by optimizing over these distributions. Gradients are then estimated via a double-loop algorithm (see details in Appendix H).

**Remark 2.** *Another possible regularization function is the squared Wasserstein distance, $\mathcal{G}(\mu) = W_2^2(\mu, v)$, which is also linearly convex in the probability space $\mu \in \mathcal{P}_2^r(\mathcal{X})$. The Wasserstein gradient of $\mathcal{L}(\mu)$ at $\mu$ at $\mu$ is given by $x - T(x)$, where $T$ denotes the optimal transport (OT) map pushing $\mu$ to $\nu$ (see Appendix G). However, computing OT is prohibitively expensive for gradient descent. By contrast, Sup-MMD offers a simple and effective alternative in practice.*

## 3.1 EXPERIMENTS

**Synthetic Data** We set the source distribution as a 2D Gaussian with mean $(0, 0)$ and unit variance, and the target as a mixture of unit-variance Gaussians centered at $(-3, -3)$, $(-3, 3)$, $(3, -3)$, and $(3, 3)$, ensuring no overlap between their high-density regions. We regularize the KL flow with Sup-MMD and vary the weight $\lambda$ in (9), also comparing against a KL-only flow trained via denoising score matching with a 4-layer MLP of 16 hidden units. Step size for Wasserstein gradient descent is set to $0.01$. Size of random Fourier features for Sup-MMD is set to $16$. We take 5000 samples from source distribution and target distribution.

In Fig. 1, the first three pictures show trajectories of 30 particles under the regularized flow with $\lambda = 0, 0.01, 0.001$, all of which transport source particles toward the target domain. The last picture shows KL flow without Sup-MMD, which fails to converge. This is because the score estimator trained on target data provides little guidance near the source data. This demonstrates that Sup-MMD supplies additional geometric information that drives particles toward target distribution. When $\lambda = 0$, (9) does not satisfy the PL condition, so convergence is not guaranteed and particles diverge (first

picture). When $\lambda$ is too large, the score term dominates in the late stage, collapsing particles to each Gaussian center (second picture). These results highlight the importance of tuning $\lambda$.

Fig. 2 plots the Sup-MMD loss for (9) with different $\lambda$. As noted in Theorem 2, $\lambda$ controls the convergence rate: larger values yield faster convergence when score estimates are accurate. In practice, however, if $\lambda$ is too large, the degenerate score estimates in the source domain dominate and convergence fails.

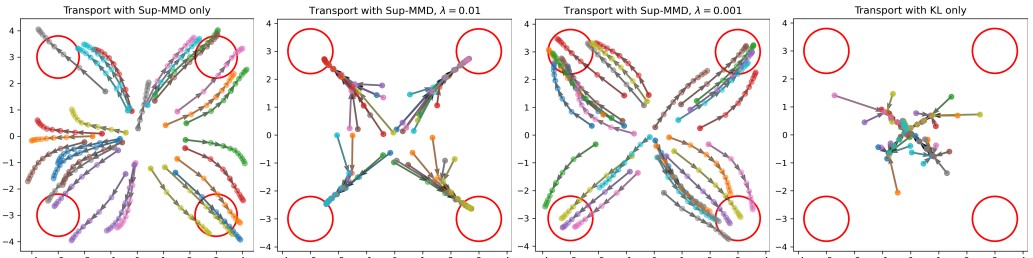

Figure 1: Particle trajectory of transporting standard Gaussian distribution to mixture gaussian centered at four corners. From left to right: Sup-MMD flow ($\lambda = 0$), (9) with $\lambda = 0.01$, (9) with $\lambda = 0.001$, KL flow.

**Image Feature Data**   We evaluate male-to-female image translation using Sup-MMD regularized KL flows on a 512-dimensional feature space extracted from a pre-trained ALAE autoencoder (Pidhorskyi et al., 2020) trained on the $1024 \times 1024$ FFHQ dataset (Karras et al., 2019). The denoising score matching network is a 4-layer MLP with 512 hidden units. We set the Wasserstein gradient descent step size to $0.01$, use 512 random Fourier features for Sup-MMD, and draw 10,000 samples from both source and target distributions.

Fig. 3 shows particle transport under different settings. The first row corresponds to KL flow with Sup-MMD ($\lambda = 0.01$ and $\lambda = 0.001$), while the last two rows show MMD-only and KL-only flows, respectively. With Sup-MMD regularization, male samples are successfully transported to the female domain, whereas KL-only flow fail. This

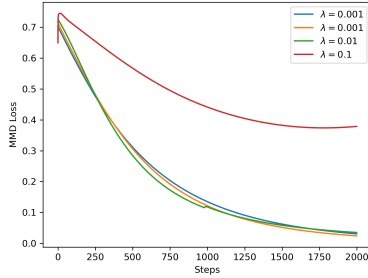

Figure 2: Sup-MMD loss for different choice of $\lambda$.

mirrors our synthetic experiments, indicating that the feature spaces of male and female images are far apart, and score estimation alone provides little guidance near the source distribution.

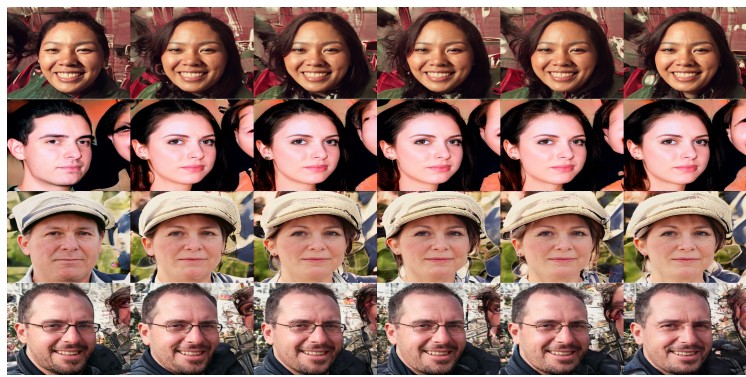

Figure 3: Trajectory of male to female translation with $\lambda = 0, 0.001, 0.01$ for 1-3 rows. KL-only flow for the last row.

## 4 APPLICATION 2: MULTI-OBJECTIVE OPTIMIZATION

**Background of multi-objective optimization**: Multi-objective optimization (MOO) problems involve optimizing multiple conflicting objectives simultaneously. These problems are prevalent in machine learning tasks such as multi-task learning Sener & Koltun (2018), neural architecture search Elsken et al. (2018), and neural combinatorial optimization Chen et al. (2023). Unlike traditional single-objective optimization problems, MOO problems do not have a single solution that can optimally satisfy all objectives. Instead, the goal is to identify a representative set of optimized solutions, known as nondominated or Pareto-optimal solutions, which form the Pareto front (PF) that represents the best achievable trade-offs in the objective space. In this context, particle-based optimization algorithms are particularly well-suited due to their ability to maintain a set of particles that approximates the PF effectively.

Commonly, a blue MOO problem can be formulated as:

$$
\begin{aligned}
\min : f(x) = \{f_1(x), \ldots, f_m(x)\}, \\
\text{s.t. } x \in \mathcal{X} \subseteq \mathbb{R}^d,
\end{aligned}
\tag{11}
$$

where $f_i(x), (i \in \{1, \ldots, m\})$ is the $i$th objective function, $m$ is the number of objectives, $x$ is the decision vector, $\mathcal{X} = \{x = (x_1, \ldots, x_d) | L_j \leq x_j \leq U_j, \ j = 1, \ldots, d\}$ is the decision space, $d$ is the dimensions of the decision vector, and $L_j$ and $U_j$ are the lower and upper bounds of the $j$th decision variable $x_j$. Some key concepts associated with the MOP are introduced as follows Deb (2011):

*Pareto Dominance*: For decision vectors $x_a$ and $x_b$, if $\forall i \in \{1, 2, \ldots, m\}$, $f_i(x_a) \leq f_i(x_b)$ and $\exists i' \in \{1, 2, \ldots, m\}$, $f_{i'}(x_a) < f_{i'}(x_b)$, $x_a$ is said to Pareto dominate $x_b$.

*Pareto Optimal Solution*: If no decision vector in $\mathcal{X}$ Pareto dominates $x_a$, then $x_a$ is a Pareto optimal solution.

*Pareto Set*: The set of all Pareto optimal solutions forms the Pareto set in decision space.

*Pareto Front*: The image of the Pareto set in the objective space forms the PF.

**SPGD for multi-objective optimization**: Building upon the work of Ren et al. (2024), we formulate the multi-objective optimization problem by defining the functional $\mathcal{L}_{moo}$ as follows:

$$
\mathcal{L}_{moo}(\mu) = \mathcal{F}(\mu) + \lambda_r \mathcal{R}(\mu) + \lambda_{ent} \text{Ent}(\mu).
\tag{12}
$$

In (12), $\mathcal{F}$ represents the objective function term designed to drive the images of particles toward the Pareto front PF, and $\lambda_r$ and $\lambda_{ent}$ are two weights. In this paper, $\mathcal{F}$ is computed as follows:

$$
\mathcal{F}(\mu) = \int_{\mathcal{X}} ||g^\dagger(x)||^2 \mu(\mathrm{d}x),
\tag{13}
$$

where

$$
-||g^\dagger(x)||^2 = \min_{||g|| \leq 1} \min_{i \in [m]} -g^\top \nabla f_i(x).
\tag{14}
$$

The calculation of (14) is detailed in Sener & Koltun (2018), and the approximation of the Wasserstein gradient of $\mathcal{F}$ is provided in Ren et al. (2024). In addition to the objective function term, a repulsive term $\mathcal{R}$ is introduced to maintain the diversity of particles in the objective space. Specifically, $\mathcal{R}$ is computed as follows:

$$
\mathcal{R}(\mu) = \frac{1}{2} \int_{\mathcal{X} \times \mathcal{X}} \mu(\mathrm{d}x) R(f(x), f(x')) \mu(\mathrm{d}x'),
\tag{15}
$$

where $R(f(x), f(x')) = \exp(\frac{||f(x) - f(x')||^2}{\sigma^2})$. Furthermore, following the suggestion of Ren et al. (2024), an entropy term is also included $\text{Ent}(\mu)$. It is worth noting that the functional $\mathcal{L}_{moo}$ defined in (12) can be interpreted as an entropy regularized objective introduced in Section 2, and we assume it satisfies the Polyak-Łojasiewicz condition. Then, by optimizing $\mathcal{L}_{moo}$ using the SPGD, the MOO problem can be solved.

Finally, we provide an extensive empirical analysis of our algorithm and compare it to the state-of-the-art WFR and MFLD algorithms in Section 4.0.1, where our approach outperforms the others.

### 4.0.1 MOO RESULTS

**MOO Benchmark Problems and Evaluation Metrics**: To evaluate the performance of SPGD, we design two benchmark problems inspired by the benchmark problems introduced in Deb et al. (2005). The designed two problems are denoted as MOO-1 and MOO-2. Specifically, MOO-1 is formulated as follows:

$$\begin{cases} f_1(x) = x_1(1 + c(x_{\mathrm{II}})), \\ f_2(x) = (1 - x_1)(1 + c(x_{\mathrm{II}})), \end{cases} \tag{16}$$

and MOO-2 is formulated as:

$$\begin{cases} f_1(x) = \sin(\frac{1}{2\pi}x_1)(1 + c(x_{\mathrm{II}})), \\ f_2(x) = \cos(\frac{1}{2\pi}x_1)(1 + c(x_{\mathrm{II}})). \end{cases} \tag{17}$$

In both (16) and (17), $x_{\mathrm{II}} = (x_2, \ldots, x_d)$, and $c(\mathbf{x}_{\mathrm{II}})$ is defined as:

$$c(\mathbf{x}_{\mathrm{II}}) = \frac{1}{b_{max}(d-1)} \sum_{b=1}^{b_{max}} \sum_{j=2}^{d} (\max\{|x_d - s_{b,d}|, 0.15\})^2. \tag{18}$$

In (18), $s_{b,d}$ represents a randomly generated bias value within the range $[0.4, 0.6]$, and $b_{max}$ is a parameter set to 320 for the purposes of this paper. MOO-1 and MOO-2 are designed to simulate the batch estimation process of the Wasserstein gradient. As shown in (16) and (17), by randomly sampling a subset $\mathcal{B} \subseteq [b_{max}]$, the Wasserstein gradient of $c(x_{\mathrm{II}})$ can be estimated as $\frac{1}{|\mathcal{B}|(d-1)} \sum_{b \in \mathcal{B}} \sum_{j=2}^{d} (\max\{|x_d - s_{b,d}|, 0.15\})^2$. For the experiments presented in this paper, we set $|\mathcal{B}|$ to 32, and set $d$ to 20. Additionally, Additionally, for both benchmark problems, the bounds $L_j$ and $U_j$ ($j \in \{1, \ldots, d\}$), are set to 0 and 1, respectively. If any decision variable falls outside this range, it is replaced with a random value within the interval $[0, 1]$.

We evaluated the performance of the algorithms using the Hypervolume (HV) metric, which quantifies the size of the region in the objective space dominated by a set of data points (Auger et al., 2012). The HV calculation requires the specification of reference points. In this study, for both MOO-1 and MOO-2, the reference points are set to $(4, 4)$.

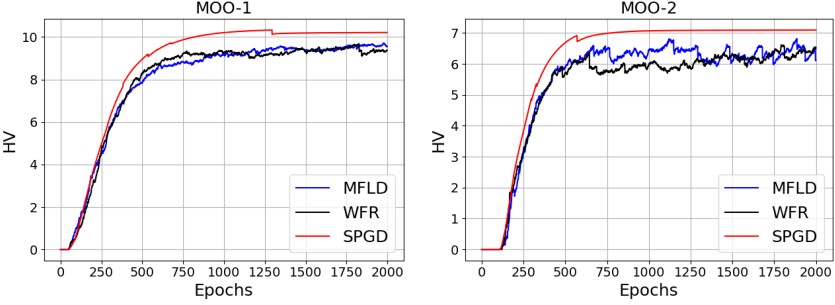

Figure 4: The HV convergence trends of MFLD, WFR, and SPGD on solving MOO-1 and MOO-2. Left: MOO-1. Right: MOO-2.

**Results**: In our experiments, we compare the performance of SPGD with two particle-based methods, namely MFLD and WFR Ren et al. (2024), for optimization. For all three algorithms, the number of particles are set to 20, the learning rate is set to 0.0001, and the parameters $\lambda_r$ and $\lambda_{ent}$ are both set to 0.01. The convergence trends of the HV metric for MFLD, WFR, and SPGD in solving MOO-1 and MOO-2 are shown in Figure 4. From the results, it can be observed that SPGD achieves faster convergence, which aligns with our theoretical analysis. Furthermore, the convergence of MFLD and WFR exhibits random fluctuations, whereas SPGD demonstrates greater stability. This difference is likely due to the fact that MFLD and WFR introduce Brownian motion to simulate the diffusion process necessary to maintain the entropy term, whereas SPGD directly optimizes the entropy term using the estimated Wasserstein gradient. To further illustrate this behavior, we present the evolution trajectories of MFLD, WFR, and SPGD in the objective space for solving MOO-1 in Fig. 5. The trajectories of MFLD and WFR show random fluctuations, while SPGD follows a more direct and

stable path. This result highlights the distinct behavioral characteristics of SPGD. Finally, we present the final approximation results of the particle distributions in MFLD, WFR, and SPGD in Figure 6. As shown, SPGD provides the best approximation of the PF for both MOO-1 and MOO-2.

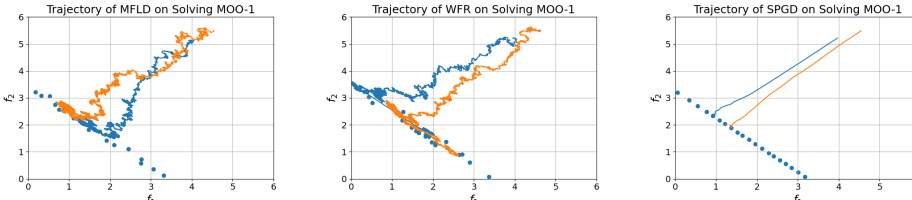

Figure 5: The evolution trajectory of MFLD, WFR, and SPGD in the objective space of MOO-1. Upper left: the evolution trajectory of MFLD on MOO-1. Upper right: the evolution trajectory of WFR on MOO-1. Bottom: The evolution trajectory of SPGD on MOO-1.

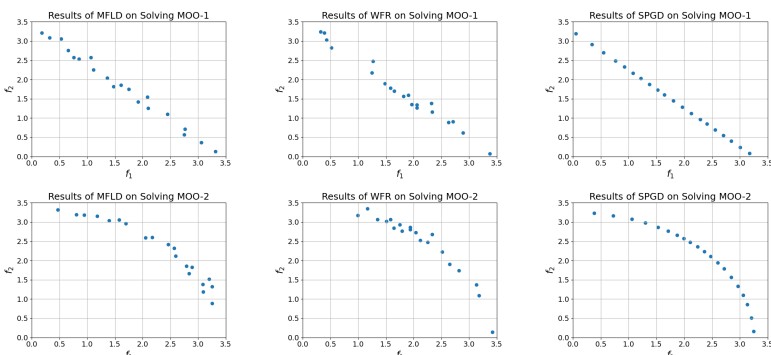

Figure 6: The PF approximation results provided by MFLD, WFR, and SPGD on solving MOO-1 and MOO-2.

In Appendix I, we present the results of multi-objective optimization (MOO) under geodesic convexity (removing the entropy penalty). The results are only slightly worse than the formulation with the entropy term, but incur significantly lower computational cost since no score function needs to be computed.

## CONCLUSION

In this work, we study deterministic transport of distribution via Wasserstein gradient flow. We establish convergence guarantees under geodesic convexity and the PL condition, based on asymptotic Wasserstein gradient estimation that accommodates biased stochastic gradient estimates. We then introduce Supremum Maximum Mean Discrepancy (Sup-MMD) as a regularizer for KL flow and demonstrate its effectiveness. Moreover, we present applications in multi-objective optimization.

A limitation of our approach lies in score estimation: when entropy terms are required for PL convexity, the estimation relies on denoising score matching, whose complexity remains insufficiently understood. Future work includes analyzing the convergence rate of Sup-MMD regularized flows and studying transport properties in applications such as image translation near the regime of optimal transport.

## REPRODUCIBILITY STATEMENT

Complete assumptions and detailed proofs of all theoretical results are provided in the main paper and appendix. Source code for reproducing all experiments is available at the anonymous link: `https://drive.google.com/drive/folders/1O30PhlcXRaZs20FzLKwD_c7nJpFxNdNn?usp=drive_link`.

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

## A   PROOF OF THEOREM 1

Assumption 1 leads to the following descent lemma.

**Lemma 1** (Descent lemma). *Under Assumption 1, we run SPGD (4) with $\eta \leq \frac{1}{4L}$. Then, we get*

$$
\mathbf{E}[\mathcal{L}(\mu_{k+1})] \leq \mathbf{E}[\mathcal{L}(\mu_k)] - \frac{\eta}{4}\mathbf{E}\left[\|\nabla_W\mathcal{L}(\mu_k)\|^2_{L_2(\mu_k;\mathbf{R}^d)}\right]
$$
$$
+ \frac{3\eta\tau}{4} + \frac{L\eta^2\sigma^2}{2}.
$$

*Proof.* By Assumption 1 and the update (5), we have

$$
\mathcal{L}(\mu_{k+1}) \leq \mathcal{L}(\mu_k) - \eta\left\langle \nabla_W\mathcal{L}(\mu_k), G_k\right\rangle_{L_2(\mu_k;\mathbf{R}^d)} + \frac{L\eta^2}{2}\|G_k\|^2_{L_2(\mu;\mathbf{R}^d)}
$$
$$
\leq \mathcal{L}(\mu_k) - \eta\|\nabla_W\mathcal{L}(\mu_k)\|^2_{L_2(\mu_k;\mathbf{R}^d)} - \eta\left\langle\nabla_W\mathcal{L}(\mu_k), G_k - \nabla_W\mathcal{L}(\mu_k)\right\rangle_{L_2(\mu_k;\mathbf{R}^d)}
$$
$$
+ \frac{L\eta^2}{2}\|G_k\|^2_{L_2(\mu;\mathbf{R}^d)}.
$$

Given $\mu_k$, we take the expectation with respect to $G_k$, leading to

$$
\mathbf{E}_{G_k}[\mathcal{L}(\mu_{k+1})]
$$
$$
\leq \mathcal{L}(\mu_k) - \eta\|\nabla_W\mathcal{L}(\mu_k)\|^2_{L_2(\mu_k;\mathbf{R}^d)} - \eta\left\langle\nabla_W\mathcal{L}(\mu_k), \mathbf{E}_{G_k}[G_k] - \nabla_W\mathcal{L}(\mu_k)\right\rangle_{L_2(\mu_k;\mathbf{R}^d)}
$$
$$
+ \frac{L\eta^2}{2}\mathbf{E}_{G_k}\left[\|G_k\|^2_{L_2(\mu_k;\mathbf{R}^d)}\right]
$$
$$
\leq \mathcal{L}(\mu_k) - \frac{\eta}{2}\|\nabla_W\mathcal{L}(\mu_k)\|^2_{L_2(\mu_k;\mathbf{R}^d)} + \frac{\eta}{2}\|\mathbf{E}[G_k] - \nabla_W\mathcal{L}(\mu_k)\|^2_{L_2(\mu_k;\mathbf{R}^d)}
$$
$$
+ \frac{L\eta^2}{2}\left(2\|\nabla_W\mathcal{L}(\mu_k)\|^2_{L_2(\mu_k;\mathbf{R}^d)} + 2\|\mathbf{E}[G_k] - \nabla_W\mathcal{L}(\mu_k)\|^2_{L_2(\mu_k;\mathbf{R}^d)}\right.
$$
$$
\left. + \mathbf{E}_{G_k}\left[\|G_k - \mathbf{E}[G_k]\|^2_{L_2(\mu_k;\mathbf{R}^d)}\right]\right).
$$
$$
\leq \mathcal{L}(\mu_k) - \frac{\eta}{2}(1 - 2L\eta)\|\nabla_W\mathcal{L}(\mu_k)\|^2_{L_2(\mu_k;\mathbf{R}^d)} + \frac{\eta}{2}(1 + 2L\eta)\tau + \frac{L\eta^2}{2}\sigma^2.
$$

This finishes the proof.  $\square$

We here provide a lemma used for showing the convergence under geodesic convexity.

**Lemma 2.** *Under Assumption 1 and 2, we run SPGD (4). Suppose $c > 0$. Then, we get*

$$
\mathbf{E}[W_2^2(\mu_{k+1}, \mu_*)] + 2\eta(\mathbf{E}[\mathcal{L}(\mu_k)] - \mathcal{L}(\mu_*))
$$
$$
\leq \left(1 - \frac{c\eta}{2}\right)\mathbf{E}[W_2^2(\mu_k, \mu_*)]
$$
$$
+ 2\eta^2\mathbf{E}\left[\|\nabla_W\mathcal{L}(\mu_k)\|^2_{L_2(\mu_k;\mathbf{R}^d)}\right] + \frac{2\eta\tau}{c} + \eta^2\sigma^2 + 2\eta^2\tau.
$$

In the case $c = 0$, the same result as Lemma 2 holds by setting $\tau = 0$.

*Proof of Lemma 2.* Let $\gamma \in \Gamma_o(\mu_k, \mu_*)$ be the optimal coupling of $\mu_k$ and $\mu_*$.

$$
W_2^2(\mu_k, \mu_*) = \int \|x - y\|_2^2\gamma(\mathrm{d}x, \mathrm{d}y)
$$
$$
= \int \|x - \eta G_k(x) + \eta G_k(x) - y\|_2^2\gamma(\mathrm{d}x, \mathrm{d}y)
$$
$$
= \int \|x - \eta G_k(x) - y\|_2^2\gamma(\mathrm{d}x, \mathrm{d}y)
$$
$$
+ 2\eta\int G_k(x)^\top(x - \eta G_k(x) - y)\gamma(\mathrm{d}x, \mathrm{d}y)
$$
$$
+ \eta^2\int \|G_k(x)\|_2^2\mu_k(\mathrm{d}x).
$$

We evaluate each term on the right-hand side as follows:

$$\int \|x - \eta G_k(x) - y\|_2^2 \gamma(\mathrm{d}x, \mathrm{d}y) = \int \|x - y\|_2^2 ((id - \eta G_k) \times id)_\sharp \gamma(\mathrm{d}x, \mathrm{d}y) \geq W_2^2(\mu_{k+1}, \mu_*),$$

and

$$\int G_k(x)^\top (x - \eta G_k(x) - y) \gamma(\mathrm{d}x, \mathrm{d}y)$$

$$= -\eta \int \|G_k(x)\|_2^2 \mu_k(\mathrm{d}x) + \int G_k(x)^\top (x - y) \gamma(\mathrm{d}x, \mathrm{d}y)$$

$$= -\eta \int \|G_k(x)\|_2^2 \mu_k(\mathrm{d}x) + \int G_k(x)^\top (x - y)(id \times t_{\mu_k}^{\mu_*})_\sharp \mu_k(\mathrm{d}x, \mathrm{d}y)$$

$$= -\eta \int \|G_k(x)\|_2^2 \mu_k(\mathrm{d}x) + \int G_k(x)^\top (x - t_{\mu_k}^{\mu_*}(x)) \mu_k(\mathrm{d}x)$$

$$= -\eta \int \|G_k(x)\|_2^2 \mu_k(\mathrm{d}x) + \int \nabla_W \mathcal{L}(\mu_k)(x)^\top (x - t_{\mu_k}^{\mu_*}(x)) \mu_k(\mathrm{d}x)$$

$$+ \int (G_k(x) - \nabla_W \mathcal{L}(\mu_k)(x))^\top (x - t_{\mu_k}^{\mu_*}(x)) \mu_k(\mathrm{d}x)$$

$$\geq -\eta \int \|G_k(x)\|_2^2 \mu_k(\mathrm{d}x) + \int (G_k(x) - \nabla_W \mathcal{L}(\mu_k)(x))^\top (x - t_{\mu_k}^{\mu_*}(x)) \mu_k(\mathrm{d}x)$$

$$+ \mathcal{L}(\mu_k) - \mathcal{L}(\mu_*) + \frac{c}{2} W_2^2(\mu_k, \mu_*).$$

Therefore, we get

$$W_2^2(\mu_k, \mu_*)$$

$$\geq \mathbf{E}_{G_k}\left[W_2^2(\mu_{k+1}, \mu_*)\right] + 2\eta(\mathcal{L}(\mu_k) - \mathcal{L}(\mu_*)) + c\eta W_2^2(\mu_k, \mu_*) - \eta^2 \mathbf{E}_{G_k}\left[\|G_k(x)\|_{L_2(\mu_k; \mathbf{R}^d)}^2\right]$$

$$+ 2\eta \int (\mathbf{E}[G_k(x)] - \nabla_W \mathcal{L}(\mu_k)(x))^\top (x - t_{\mu_k}^{\mu_*}(x)) \mu_k(\mathrm{d}x)$$

$$\geq \mathbf{E}_{G_k}\left[W_2^2(\mu_{k+1}, \mu_*)\right] + 2\eta(\mathcal{L}(\mu_k) - \mathcal{L}(\mu_*)) + c\eta W_2^2(\mu_k, \mu_*) - \eta^2 \mathbf{E}_{G_k}\left[\|G_k(x)\|_{L_2(\mu_k; \mathbf{R}^d)}^2\right]$$

$$- \eta \left(\frac{2}{c} \|\mathbf{E}[G_k(x)] - \nabla_W \mathcal{L}(\mu_k)(x)\|_{L_2(\mu_k; \mathbf{R}^d)}^2 + \frac{c}{2} W_2^2(\mu_k, \mu_*)\right)$$

$$\geq \mathbf{E}_{G_k}\left[W_2^2(\mu_{k+1}, \mu_*)\right] + 2\eta(\mathcal{L}(\mu_k) - \mathcal{L}(\mu_*)) + \frac{c\eta}{2} W_2^2(\mu_k, \mu_*) - \eta^2 \mathbf{E}_{G_k}\left[\|G_k(x)\|_{L_2(\mu_k; \mathbf{R}^d)}^2\right]$$

$$- \frac{2\tau\eta}{c}$$

Noticing $\mathbf{E}_{G_k}[\|G_k\|_{L_2(\mu_k; \mathbf{R}^d)}^2] \leq \sigma^2 + 2\tau + 2\|\nabla_W \mathcal{L}(\mu_k)\|_{L_2(\mu_k; \mathbf{R}^d)}^2$ by Assumption 1, we finish the proof. $\qquad \square$

Next, we provide the proof of Theorem 1. By Lemma 1 and 2, we see

$$\mathbf{E}[W_2^2(\mu_{k+1}, \mu_*)] + 2\eta(\mathbf{E}[\mathcal{L}(\mu_k)] - \mathcal{L}(\mu_*))$$

$$\leq \left(1 - \frac{c\eta}{2}\right) \mathbf{E}[W_2^2(\mu_k, \mu_*)] + 8\eta(\mathbf{E}[\mathcal{L}(\mu_k)] - \mathbf{E}[\mathcal{L}(\mu_{k+1})]) + \frac{2\eta\tau}{c} + 8\eta^2\tau + 2\eta^2\sigma^2. \quad (19)$$

We first consider the case: $c = \tau = 0$, where the same inequality holds. Hence, by summing over $k \in \{0, \ldots, T-1\}$, we get

$$\mathbf{E}[W_2^2(\mu_T, \mu_*)] - W_2^2(\mu_0, \mu_*) + 2\eta \sum_{k=0}^{T-1} (\mathbf{E}[\mathcal{L}(\mu_k)] - \mathcal{L}(\mu_*))$$

$$\leq 8\eta(\mathcal{L}(\mu_0) - \mathbf{E}[\mathcal{L}(\mu_T)]) + 2\eta^2\sigma^2 T.$$

This proves the first statement:

$$\frac{1}{T} \sum_{k=0}^{T-1} (\mathbf{E}[\mathcal{L}(\mu_k)] - \mathcal{L}(\mu_*)) \leq \frac{1}{2T} \left(\frac{1}{\eta} W_2^2(\mu_0, \mu_*) + 8\mathcal{L}(\mu_0)\right) + \eta\sigma^2.$$

We next prove the second statement. Since $6\eta \leq 8\eta(1 - c\eta/2)$, Eq. (19) leads to

$$\mathbf{E}[W_2^2(\mu_{k+1}, \mu_*)] + 8\eta(\mathbf{E}[\mathcal{L}(\mu_{k+1})] - \mathcal{L}(\mu_*))$$

$$\leq \left(1 - \frac{c\eta}{2}\right) \mathbf{E}[W_2^2(\mu_k, \mu_*)] + 6\eta(\mathbf{E}[\mathcal{L}(\mu_k)] - \mathcal{L}(\mu_*)) + \frac{2\eta\tau}{c} + 8\eta^2\tau + 2\eta^2\sigma^2$$

$$\leq \left(1 - \frac{c\eta}{2}\right) \left(\mathbf{E}[W_2^2(\mu_k, \mu_*)] + 8\eta(\mathbf{E}[\mathcal{L}(\mu_k)] - \mathcal{L}(\mu_*))\right) + \frac{2\eta\tau}{c} + 8\eta^2\tau + 2\eta^2\sigma^2.$$

We set $V_k = \mathbf{E}[W_2^2(\mu_k, \mu_*)] + 8\eta(\mathbf{E}[\mathcal{L}(\mu_k)] - \mathcal{L}(\mu_*))$. Then, this inequality is equivalent to

$$V_{k+1} - \frac{4\tau}{c^2} - \frac{16\eta\tau}{c} - \frac{4\eta\sigma^2}{c} \leq \left(1 - \frac{c\eta}{2}\right) \left(V_k - \frac{4\tau}{c^2} - \frac{16\eta\tau}{c} - \frac{4\eta\sigma^2}{c}\right).$$

By recursively applying this, we get

$$V_T \leq \frac{4\tau}{c^2} + \frac{16\eta\tau}{c} + \frac{4\eta\sigma^2}{c} + \left(1 - \frac{c\eta}{2}\right)^T V_0,$$

which finishes the proof.

# B  PROOF OF EXAMPLES OF PL CONDITION

Given a linearly convex functional $F : \mathcal{P}_2(\mathbf{R}^d) \to \mathbf{R}$, we consider the entropy regularized objective function: for $\mu \in \mathcal{P}_2(\mathbf{R}^d)$,

$$\mathcal{L}(\mu) = F(\mu) + \lambda' \mathbf{E}_{X \sim \mu}[\|X\|_2^2] + \lambda \mathrm{Ent}(\mu). \tag{20}$$

We suppose that there exists $B > $ such that for any $\mu \in \mathcal{P}_2^r(\mathbf{R}^d)$, $\|\frac{\delta F(\mu)}{\delta \mu}\|_\infty \leq B$. Then, by Holley and Stroock argument (Holley & Stroock, 1987), log-Sobolev inequality (LSI) with the constant $\alpha = \frac{2\lambda'}{\lambda \exp(4B/\lambda)}$ holds:

$$\mathrm{KL}(\mu \| \hat{\mu}) \leq \frac{1}{2\alpha} \mathbf{E}_{X \sim \mu} \left[\left\|\nabla \log \frac{\mathrm{d}\mu}{\mathrm{d}\hat{\mu}}(X)\right\|_2^2\right], \tag{21}$$

where $\hat{\mu}$ is the proximal Gibbs distribution defined by $\frac{\mathrm{d}\hat{\mu}}{\mathrm{d}x}(x) \propto \exp\left(-\frac{1}{\lambda} \frac{\partial F}{\partial \mu}(x) - \frac{\lambda'}{\lambda} \|x\|_2^2\right)$. The right hand side of LSI (21) is same as $\frac{1}{2\alpha\lambda^2} \|\nabla_W \mathcal{L}(\mu)\|_{L_2(\mu;\mathbf{R}^d)}^2$. Applying the entropy sandwich (Nitanda et al., 2022; Chizat, 2022): $\mathcal{L}(\mu) - \mathcal{L}(\mu_*) \leq \lambda \mathrm{KL}(\mu \| \hat{\mu})$, we deduce $\mathcal{L}(\mu) - \mathcal{L}(\mu_*) \leq \frac{1}{2\alpha\lambda} \|\nabla_W \mathcal{L}(\mu)\|_{L_2(\mu;\mathbf{R}^d)}^2$.

# C  CONVERGENCE PROOF UNDER PL CONDITION

*Proof.* By Lemma 1 and Assumption 3, we get

$$\mathbf{E}[\mathcal{L}(\mu_{k+1})] - \mathcal{L}(\mu_*)$$

$$\leq \left(1 - \frac{\eta c}{4}\right) (\mathbf{E}[\mathcal{L}(\mu_k)] - \mathcal{L}(\mu_*)) + \frac{3\eta\tau}{4} + \frac{L\eta^2\sigma^2}{2}.$$

This is equivalent to

$$\mathbf{E}[\mathcal{L}(\mu_{k+1})] - \mathcal{L}(\mu_*) - \frac{3\tau}{c} - \frac{2L\eta\sigma^2}{c}$$

$$\leq \left(1 - \frac{\eta c}{4}\right) \left(\mathbf{E}[\mathcal{L}(\mu_k)] - \mathcal{L}(\mu_*) - \frac{3\tau}{c} - \frac{2L\eta\sigma^2}{c}\right).$$

Hence, we conclude

$$\mathbf{E}[\mathcal{L}(\mu_T)] - \mathcal{L}(\mu_*)$$

$$\leq \left(1 - \frac{\eta c}{4}\right)^T (\mathcal{L}(\mu_0) - \mathcal{L}(\mu_*)) + \frac{3\tau}{c} + \frac{2L\eta\sigma^2}{c}.$$

$\square$

# D  BACKGROUND OF SPGD

Distribution optimization considers optimizing functionals defined over the space of probability measures. This setup has various applications, including generative modeling (Arjovsky et al., 2017), training neural network (Nitanda & Suzuki, 2017; Mei et al., 2018; Chizat & Bach, 2018), ensemble learning (Nitanda & Suzuki, 2017), Bayesian inference (Welling & Teh, 2011; Dai et al., 2016; Liu & Wang, 2016), and distribution matching (Arbel et al., 2019).

Optimization over measure space inevitably comes with sophistication in methodology that depends on the complexity and geometry of measures, which goes beyond traditional optimization framework in Euclidean space. Various methods have been developed based on different principles, such as variational technique (Welling & Teh, 2011; Liu & Wang, 2016; Wang & Liu, 2019), proximal algorithm (Jordan et al., 1998; Lascu et al., 2024), and mean-field Langevin dynamics (Mei et al., 2018; Hu et al., 2021; Nitanda et al., 2022; Chizat, 2022). In particular, *Wasserstein gradient flow* (WGF) (Ambrosio et al., 2008) is the most basic framework that extends the concept of steepest descent from Euclidean to the space of measures endowed with the Wasserstein metric, providing a potentially efficient framework for distribution optimization with fast convergence. In fact, many distribution optimization methods introduced above build upon the concept of WGF.

Various methods have been developed for distribution optimization in terms of different principles and levels of sophistication. Langevin dynamics can be considered as a distribution optimization method for minimizing KL divergence (Jordan et al., 1998), and often used as a sampling method in many contexts such as Bayesian inference (Welling & Teh, 2011). Liu & Wang (2016) also proposed Stein Variational Gradient Descent (SVGD) to optimize KL divergence with kernelization. Wang & Liu (2019) extended this method to optimize nonlinear objectives with entropy regularization. There are also variations of SVGD, such as using perturbed particles (Zhang et al., 2024) or parameterizing mixture models (Rønning et al., 2024). Cheng et al. (2023) studied generalized Wasserstein gradient flow for the KL divergence with a focus on theoretical convergence.

Wasserstein proximal algorithm (Jordan et al., 1998) is a time-discretization of continuous-time WGF. Salim et al. (2020) also proposed a discretization method for objectives with geodesically convex terms, giving convergence guarantees similar to proximal gradient algorithms in Euclidean space. Lascu et al. (2024) showed linear convergence under a uniform logarithmic Sobolev inequality (Nitanda et al., 2022; Chizat, 2022) for entropy-regularized linearly convex functional. Yao et al. (2024) explored geodesically convex optimization over multiple distributions by combining convex potential energy with self-interaction and internal energies.

Mean-field Langevin dynamics (MFLD) (Mei et al., 2018; Hu et al., 2021) minimizes an entropy-regularized linearly convex functional. Chen et al. (2024) studied particle approximation errors based on the uniform-in-time propagation of chaos. This result inspired several subsequent works such as discretization analysis (Suzuki et al., 2023), sampling from mean-field stationary distribution (Kook et al., 2024), and refinement with respect to particle approximation error (Nitanda, 2024). MFLD works for minimizing linearly convex functional with the entropy regularization. In the MFLD, the entropy term is simulated by using the Brownian motion whereas our method directly represents the entropy regularization through the Wasserstein gradient estimate. This makes the analysis simpler and improves theoretical guarantees by removing the dependency on the entropy term. It also allows us to update particles as a pushforward measure, which avoids the difficulties of analyzing randomness in MFLD.

The typical conditions on $\mathcal{F}$ used in the literature to show the convergence are (i) *geodesic convexity (displacement convexity)* under which the functional is convex along Wasserstain geodesic and (ii) *linear convexity (flat convexity)* under which the functional is convex along linear interpolation of any two distributions. For instance, the convergence analysis of WGF under geodesic convexity can be found in Ambrosio et al. (2008), and this convexity has been exploited in several methods (Arbel et al., 2019; Daneshmand et al., 2023). Recently, the linear convexity has drawn more attention because of its connection to the training neural networks. Nitanda & Suzuki (2017) translated the stochastic gradient descent for neural networks under the mean-field regime to the distribution optimization methods referred to as *stochastic particle gradient descent* (SPGD) for optimizing linear convex functional. SPGD is also time-and space-discretization of WGF using finite-particle systems $\frac{1}{N}\sum_{i=1}^{N}\delta_{x^i}, (x^i \in \mathbb{R}^d)$. Afterwards, Chizat & Bach (2018); Mei et al. (2018) proved the global convergence to the solution under the mean-field limit $N \to \infty$. Meanwhile, Nitanda et al.

(2022); Chizat (2022) studied the convergence rate of mean-field Langevin dynamics (MFLD), noisy gradient descent for minimizing linearly convex functional plus negative entropy regularization. Their objective function falls into a more general class of *Polyak-Łojasiewicz* (PL) condition. This problem class also includes Wasserstein barycenter estimation (Chewi et al., 2020; Chizat, 2023; Backhoff et al., 2024). More recently, Zhu & Chen (2025) established the convergence of Wasserstein proximal algorithm under both geodesic convexity and PL-condition.

Our work provides theoretical convergence guarantees for SPGD under both geodesic convexity and the PL condition. Notably, our framework allows for bias in the Wasserstein gradient estimator, which further broadens its applicability.

# E    PROOF OF CONVEXITY OF IPM

*Proof.* Let $\mu_1$ and $\mu_2$ be probability measures in $\mathcal{P}_2^r(\mathcal{X})$ and let $t \in [0, 1]$. Set $\mu_t := t\mu_1 + (1-t)\mu_2$. For any fixed $f \in \mathcal{F}$,

$$\mathbb{E}_{\mu_t}[f] - \mathbb{E}_{\nu}[f] = t\left(\mathbb{E}_{\mu_1}[f] - \mathbb{E}_{\nu}[f]\right) + (1-t)\left(\mathbb{E}_{\mu_2}[f] - \mathbb{E}_{\nu}[f]\right).$$

Taking absolute values and using convexity of $|\cdot|$,

$$|\mathbb{E}_{\mu_t}[f] - \mathbb{E}_{\nu}[f]| \le t|(\mathbb{E}_{\mu_1}[f] - \mathbb{E}_{\nu}[f])| + (1-t)|(\mathbb{E}_{\mu_2}[f] - \mathbb{E}_{\nu}[f])|.$$

Now take sup over $f \in \mathcal{F}$ on the left and right:

$$\mathrm{IPM}_{\mathcal{F}}(\mu_t, \nu) = \sup_{f \in \mathcal{F}}|\mathbb{E}_{\mu_t}[f] - \mathbb{E}_{\nu}[f]| \le t\,\mathrm{IPM}_{\mathcal{F}}(\mu_1, \nu) + (1-t)\,\mathrm{IPM}_{\mathcal{F}}(\mu_2, \nu).$$

This proves convexity of $\mathcal{L}(\mu) = \mathrm{IPM}_{\mathcal{F}}(\mu, \nu)$. $\qquad\square$

# F    WASSERSTEIN GRADIENT OF IPM

*Proof.* Let $\nu$ be a fixed probability measure and let

$$\mathcal{L}(\mu) = \sup_{f \in \mathcal{F}}\left\{\int f\,\mathrm{d}\mu - \int f\,\mathrm{d}\nu\right\}.$$

Consider a smooth compactly supported vector field $\psi$ and form the push-forward perturbation $\mu_\epsilon = (\mathrm{id} + \epsilon\psi)_{\#}\mu$. For any fixed $f \in \mathcal{F}$, we have

$$\int f\,\mathrm{d}\mu_\epsilon = \int f\big(x + \epsilon\psi(x)\big)\,\mathrm{d}\mu(x) = \int \left(f(x) + \epsilon\langle\nabla f(x), \psi(x)\rangle\right)\,\mathrm{d}\mu(x) + o(\epsilon).$$

Therefore,

$$\frac{\mathrm{d}}{\mathrm{d}\epsilon}\Big|_{\epsilon=0}\int f\,\mathrm{d}\mu_\epsilon = \int \langle\nabla f(x), \psi(x)\rangle\,\mathrm{d}\mu(x).$$

By definition of $\mathcal{L}$,

$$\mathcal{L}(\mu_\epsilon) = \sup_{f \in \mathcal{F}}\left\{\int f\,\mathrm{d}\mu_\epsilon - \int f\,\mathrm{d}\nu\right\}.$$

By the envelope theorem and since the supremum is attained at $f_\mu$ for $\epsilon = 0$, differentiating the supremum at $\epsilon = 0$ yields

$$\frac{\mathrm{d}}{\mathrm{d}\epsilon}\Big|_{\epsilon=0}\mathcal{L}(\mu_\epsilon) = \frac{\mathrm{d}}{\mathrm{d}\epsilon}\Big|_{\epsilon=0}\left(\int f_\mu\,\mathrm{d}\mu_\epsilon - \int f_\mu\,\mathrm{d}\nu\right) = \int \langle\nabla f_\mu(x), \psi(x)\rangle\,\mathrm{d}\mu(x).$$

The proof is complete by the definition of the Wasserstein gradient (the Riesz representation of the first variation in the tangent inner product). $\qquad\square$

## G  WASSERSTEIN GRADIENT OF WASSERTEIN-2 DISTANCE

*Proof.* Let $T : \mathbb{R}^d \to \mathbb{R}^d$ with $T_{\#}\mu = \nu$ be the optimal transport map:

$$\mathcal{L}(\mu) = W_2^2(\mu, \nu) = \int \|x - T(x)\|^2 \, \mathrm{d}\mu(x).$$

Let $\psi$ a continuous and bounded function, and consider the perturbation

$$\mu_\epsilon = (\mathrm{id} + \epsilon\psi)_{\#}\mu, \qquad \epsilon \text{small}.$$

Let $T_\epsilon$ an optimal map sending $\mu_\epsilon$ to $\nu$. By optimality of $T_\epsilon$,

$$W_2^2(\mu_\epsilon, \nu) = \int \|z - T_\epsilon(z)\|^2 \, \mathrm{d}\mu_\epsilon(z) \leq \int \|(\mathrm{id} + \epsilon\psi)(x) - T(x)\|^2 \, \mathrm{d}\mu(x),$$

because transporting $z = (\mathrm{id} + \epsilon\psi)(x)$ to $T(x)$ is a coupling between $\mu_\epsilon$ and $\nu$. Expanding the right-hand side in $\varepsilon$ gives

$$\mathcal{L}(\mu_\epsilon) \leq \mathcal{L}(\mu) + \epsilon \int \langle x - T(x), \psi(x) \rangle \, \mathrm{d}\mu(x) + o(\epsilon).$$

Hence the one-sided derivative satisfies

$$\limsup_{\epsilon \downarrow 0} \frac{\mathcal{L}(\mu_\epsilon) - \mathcal{L}(\mu)}{\epsilon} \leq \int \langle x - T(x), \psi(x) \rangle \, \mathrm{d}\mu(x).$$

To obtain the reverse inequality, run the same argument starting from $\mu$ and comparing with the map $T_\epsilon \circ (\mathrm{id} + \epsilon\psi)^{-1}$ as an admissible transport from $\mu$ to $\nu$, or equivalently apply the previous inequality with $-\psi$ and $\mu_\epsilon$ in place of $\mu$. Combining the two inequalities yields

$$\frac{\mathrm{d}}{\mathrm{d}\epsilon}\Big|_{\epsilon=0} \mathcal{L}(\mu_\epsilon) = \int \langle x - T(x), \psi(x) \rangle \, \mathrm{d}\mu(x).$$

The proof is complete by the definition of the Wasserstein gradient (the Riesz representation of the first variation in the tangent inner product). $\qquad\square$

## H  SUP-MMD UNDER RANDOM FOURIER FEATURE

Random Fourier Feature (RFF) is an effective technique for scaling up kernel methods. The underlying principle of RFF is the consequence of Botcher's theorem (Bochner, 1932), which guarantees that any bounded, continuous and shift-invariant kernel, if properly scaled, is the Fourier transform of a probability measure (which is typically called the spectral measure of the kernel).

**Theorem 3.** *A continuous kernel $k(x, y) = k(x - y)$ on $\mathbb{R}^d$ is positive definite if and only if $k(\delta)$ is the Fourier transform of a non-negative measure:*

$$k(x, y) = \int_{\mathcal{E}} \mathrm{e}^{-2\pi i\omega(x-y)} \mathrm{d}\tau(\omega)$$

$$= \int_{\mathcal{E}} \left(\mathrm{e}^{-2\pi i\omega x}\right) \left(\mathrm{e}^{-2\pi i\omega y}\right)^* \mathrm{d}\tau(\omega),$$

where $i$ here denotes the imaginary unit, and $(\cdot)^*$ denotes the conjugate and $\tau(\cdot)$ is a measure on $\mathcal{E}$.

For real-valued kernel, we can ignore the imaginary part in this equation and hence the integral above converges when the complex exponentials are replaced with cosines. As a result, we obtain a real-valued mapping that satisfies the condition $k(x, y) = \mathbb{E}_{\omega,\theta}[u(x)u(y)]$ by setting $u(x) = \sqrt{2}\cos(\omega x + \theta)$ where $\omega$ follows distribution $\tau$ and $\theta$ is uniformly distributed in $[0, 2\pi]$.

The main idea behind RFF is to approximate the kernel function using its Monte-Carlo estimate:

$$k(x, y) \approx \tilde{k}(x, y) = \mathbf{u}(x)^{\mathsf{T}}\mathbf{u}(y),$$

where $\mathbf{u}(x) \in \mathbb{R}^l$ is a finite-dimensional random column vector:

$$\mathbf{u}(x) = \sqrt{\frac{2}{l}} \left[ \cos(\omega_1 x + \theta_1), \ldots, \cos(\omega_l x + \theta_l) \right]^{\mathsf{T}},$$

with $\omega_1, \ldots, \omega_l$ being i.i.d. random variables following distribution $\tau$ and $\theta_1, \ldots, \theta_l$ being uniformly distributed in $[0, 2\pi]$.

Therefore, the kernel inner product in the infinite dimensional feature space can be approximated by a pointwise product in a finite dimensional space: $\langle \phi(x), \phi(y) \rangle_{\mathcal{H}} \approx \mathbf{u}(x)^{\mathsf{T}} \mathbf{u}(y)$.

The MMD under RFF can be expressed as

$$\mathrm{MMD}_{\omega \sim \tau}(\mu, \nu) = \|\mathbb{E}_{x \sim \mu}[\mathbf{u}(x)] - \mathbb{E}_{y \sim \nu}[\mathbf{u}(y)]\|,$$

where $\|\cdot\|$ denotes the Euclidean norm. Now the distributions $\tau \in \mathcal{P}_2(\mathbb{R}^d)$ form the class of bounded kernels and $\mathrm{MMD}_{\omega \sim \tau}(\mu, \nu)$ is a linear functional w.r.t. distribution $\tau$. This yields a double-loop SPGD for Sup-MMD:

---

**Algorithm 1** SPGD of Sup-MMD under RFF

---

**Input:** Samples from source distribution $\mu$ and target distribution $\nu$.
**Initialize:** Initialize particles of $\mu_1$ using samples from source distribution $\mu$. Initialize particles of RFF frequency distribution $\tau$ using Gaussian samples.
**for** $k = 1$ **to** $T$ **do**
  Run SPGD to solve $\min_{\tau \in \mathcal{P}_2(\mathbb{R}^d)} \mathrm{MMD}_{\omega \sim \tau}(\mu_k, \nu)$.
  Compute the estimated Wasserstein gradient, and update the particles of $\mu_k$.
**end for**

---

## I  MOO under Geodesic Convexity

In Section 4, we presented the results of SPGD for solving MOO problems under the PL condition, obtained by solving (12). By removing the entropy term $\mathrm{Ent}(\mu)$ from (12), we obtain another functional that can also be used to address MOO:

$$\mathcal{L}_{\mathrm{moo}}(\mu) = \mathcal{F}(\mu) + \lambda_r \mathcal{R}(\mu). \tag{22}$$

This formulation satisfies geodesic convexity, and the proposed SPGD method can be directly applied to solve it. To evaluate the effectiveness of SPGD under geodesic convexity, we tested it on MOO-1 and MOO-2, denoted as SPGD-wo-Ent, with the same parameter settings as in the previous experiments. The convergence trends of the HV metric for SPGD-wo-Ent on MOO-1 and MOO-2 are reported in Fig. (7). For comparison, we also include the convergence trends obtained by SPGD when solving (12). The results demonstrate that, under geodesic convexity, SPGD still achieves competitive convergence.

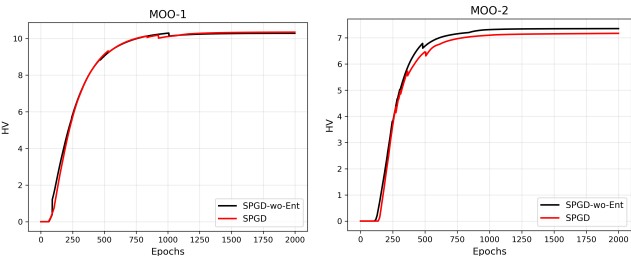

Figure 7: The HV convergence trends of SPGD and SPGD-wo-Ent on solving MOO-1 and MOO-2. Left: MOO-1. Right: MOO-2.

## J    LLM USAGE STATEMENT

Large Language Models (LLMs) were used solely to improve the clarity and readability of the writing. No content, experimental results, or research ideas were generated by LLMs.

