# OpenReview forum: "Deterministic Transport-Based Sampling via Wasserstein Gradient Flows"
_ICLR.cc/2026/Conference — Submitted to ICLR 2026_

### Official Review · Reviewer_u6LU · 2025-10-30

**Soundness:** 3
**Presentation:** 3
**Contribution:** 3
**Rating:** 6
**Confidence:** 4

**Summary:**

This paper studies deterministic transport-based sampling by formulating sampling as a Wasserstein Gradient Flow (WGF) over probability measures. The authors introduce Stochastic Particle Gradient Descent (SPGD), a particle-based discretization of WGF that generalizes score-based transport (e.g., diffusion models) and accommodates biased gradient estimates.

Three main contributions stand out: (i) The convergence under geodesic convexity and Polyak–Łojasiewicz (PL) conditions, with explicit rates that depend on the mean and variance of the gradient bias; (ii) A supremum maximum mean discrepancy (Sup-MMD) regularizer that stabilizes KL flows by enriching geometry beyond information geometry; (iii) A deterministic multi-objective optimization framework where SPGD efficiently finds Pareto-optimal distributions, outperforming WFR and MFLD methods in convergence and stability

**Strengths:**

1. The paper provides clean convergence analyses of SPGD under both geodesic convexity and PL conditions, with explicit dependence on bias and variance in the gradient estimation. This generalizes prior work on Wasserstein Langevin and mirror descent by allowing biased stochastic approximations of the Wasserstein gradient;

2. By decoupling from Langevin-type stochasticity, the proposed deterministic transport flows offer smoother trajectories and eliminate noise accumulation—a theoretically interesting and practically stable alternative to diffusion models;

3. The introduction of Sup-MMD as an adaptive geometry-aware regularizer is insightful. It remedies failures of pure KL flows when source and target distributions have disjoint supports, leading to better particle transport and convergence;

4. Experiments on synthetic Gaussian mixtures and real image features convincingly demonstrate the stability and effect of Sup-MMD. The MOO experiments show strong empirical evidence that SPGD converges faster and more stably than stochastic counterparts like MFLD and WFR;

5. The paper situates its contributions neatly in the context of Wasserstein optimization, bridging deterministic transport and variational inference frameworks. The presentation and proofs are mathematically rigorous and readable.

**Weaknesses:**

1. While SPGD is compared to MFLD and WFR, the paper lacks benchmarking against score-based deterministic samplers such as Conditional Flow Matching (CFM) or Score-based Deterministic Transport (Ilin et al., 2025). This makes it difficult to assess whether SPGD offers practical benefits in generative tasks beyond theoretical elegance.

2. Computing the Wasserstein gradient with IPM regularization or Sup-MMD requires critic optimization or kernel summations, which may not scale to high-dimensional datasets.

3. Experiments, though illustrative, remain small-scale (2D synthetic and low-dimensional FFHQ features). There is no quantitative comparison on large generative datasets or continuous-time flows.

4. The convergence proofs rely on geodesic convexity and PL conditions, which are restrictive and rarely hold in realistic generative modeling tasks.

5. The paper introduces multiple components (SPGD, Sup-MMD, entropy regularization) but does not isolate their contributions clearly—especially the role of $\lambda$ and kernel choices in convergence.

**Questions:**

1. How does SPGD compare to deterministic score-based samplers such as Conditional Flow Matching (Tong et al., 2023) or Score-based Deterministic Sampling (Ilin et al., 2025)?
2. Can Sup-MMD be efficiently implemented in high-dimensional latent spaces without large memory overhead?
3. How sensitive is the convergence to the bias/variance parameters or step size?
4. Could the framework be extended to time-dependent objectives for dynamic targets?

**Details Of Ethics Concerns:**

N/A.

---

> ### Comment · Reviewer_u6LU · 2025-11-27
> **No Rebuttal**
>
> After reading the comments from other reviewers. I decide to lower my score to 4.

---

### Official Review · Reviewer_Qgbt · 2025-10-31

**Soundness:** 3
**Presentation:** 2
**Contribution:** 3
**Rating:** 2
**Confidence:** 4

**Summary:**

This paper studies the problem of sampling from a target distribution that is implicitly defined through an optimization problem. To solve this problem, the authors develop a stochastic Wasserstein gradient flow framework that transports particles from an initial distribution to the target distribution. Under geodesic convexity and a PL inequality, convergence results are given in expected loss and expected Wasserstein distance. Two applications of flow with IPM regularization and multi-objective optimization are given, where the authors demonstrate improved performance.

**Strengths:**

- The paper develops guarantees for Wasserstein stochastic gradient descent in geodesically convex and PL inequality settings.
- Experiments demonstrate the advantages of regularized formulations as well as the advantages of the stochastic gradient method in some settings. I think that the multi-objective optimization is particularly cool, and SPGD seems to have an advantage here.
- The theoretical setup is easy to understand and the theorem statements are clear.

**Weaknesses:**

- The optimization results seem like a standard rehashing of results from convex/PL inequality-based optimization.
- The experiments are a bit light. Not many are run, and there are not many insights to gain other than that it can converge faster than other methods in these two settings.
- Theoretical results are only given in the infinite particle setting.
- The authors could do a better job of motivating the problem they are studying up front, rather than leaving motivating problems to the last two sections.
- It is hard to get a sense of the contributions of this paper, as the authors don't clearly state them up front.

**Questions:**

- Is it possible to use Flow Matching/Rectified Flows to solve these optimization problems and learn trajectories from initialization to target minimizer?
- Can the authors do a full review of related theoretical work, such as Lanzetti et al '23, to see how their analysis differs?

References:
Liu, Xingchao, and Chengyue Gong. "Flow Straight and Fast: Learning to Generate and Transfer Data with Rectified Flow." The Eleventh International Conference on Learning Representations. (2023)
Lipman, Yaron, et al. "Flow Matching for Generative Modeling." The Eleventh International Conference on Learning Representations. (2022)
Lanzetti, Nicolas, et al. "Stochastic Wasserstein gradient flows using streaming data with an application in predictive maintenance." IFAC-PapersOnLine 56.2 (2023): 3954-3959.

---

### Official Review · Reviewer_QB2w · 2025-10-31

**Soundness:** 2
**Presentation:** 2
**Contribution:** 2
**Rating:** 2
**Confidence:** 3

**Summary:**

The paper investigates deterministic sampling from target distributions implicitly defined as minimizers of objective functionals, employing the Wasserstein Gradient Flow (WGF) framework to minimize such functionals via particle-based gradient approximations. The authors establish convergence guarantees under geodesic convexity and the Polyak–Łojasiewicz (PL) condition, assuming a biased Wasserstein gradient estimation model. They further demonstrate the practical relevance of their approach in two applications:

1. Introducing the Supremum Maximum Mean Discrepancy (Sup-MMD) as a regularizer for KL flows, evaluated on an image translation task; and

2. Applying the framework to multi-objective optimization problems.

**Strengths:**

The authors address the problem of transport-based sampling for target distributions implicitly defined as minimizers of objective functionals over probability measures. These measures are approximated using an infinite-particle representation. A natural assumption is made regarding the bias in gradient evaluations, which arise from particle-based estimations of the objective. Under this setting, the authors derive convergence guarantees for both geodesically convex objectives and those satisfying the Polyak–Łojasiewicz (PL) condition.

To overcome the limitations of modeling the Wasserstein Gradient Flow of the $\mathrm{KL}(\mu \Vert \nu)$ divergence, where $\nu$ is the target distribution, via score-based methods (which provide limited guidance for particles initialized under $\mu_0$), the authors proposed Integral Probability Metrics (IPMs) as regularizers for $\mathrm{KL}$ objective. They demonstrate that this regularization improves performance in both synthetic and image-based experiments.

**Weaknesses:**

---

### **Overall Clarity and Structure**

The paper suffers from low overall clarity. The main ideas are difficult to follow, and the exposition appears inconsistent. Several sections include non-essential material in the main text, such as the plots in Section 4, that would be more appropriately placed in the Appendix. In addition, the paper’s overall structure is unconventional and does not follow standard presentation norms, e.g., no related work section in the main text, conclusion section is not numerated, background is merged with method explanation. These complicates the understanding of the paper.

---

### **References and Related Work**

Section 2 contains only a few references and provides insufficient explanation of several non-trivial concepts. For example, it is unclear why Definition (1) of the Wasserstein Gradient Flow is equivalent to the classical formulation in terms of measures (see Definition 11.1.1 in [1]). Similarly, the statement that “SPGD can be seen as a stochastic optimization method that discretizes the WGF in $\mathcal{P}_2(\mathbb{R}^d)$" is conceptually non-trivial and should be either properly cited or explained in greater detail.

The related work section is placed entirely in Appendix D without any indication in the main text, despite the paper not reaching the page limit. It would be preferable to include a concise summary of the most relevant prior work in the main body and refer readers to the appendix for an extended discussion.

Moreover, the paper does not cite existing neural network–based approaches for minimizing functionals over distribution spaces, such as [1] and [2].

---
### **Theoretical Section**

The authors state that they “study the convergence capability of SPGD (4) in the infinite-particle setting.” However, they do not justify whether this assumption is realistic or clarify for which finite $N$ this setting remains a valid approximation.

Assumption 2 defines a $c$-geodesically convex function, yet Example 1 presents only geodesically convex functions. This inconsistency may confuse readers and should be clarified.

---
### **Practical Aspects and Experiments**

The proposed algorithm is actually a minimax procedure. However, the authors do not clearly reveal this fact, giving the impression that they are downplaying the fact that such an objective can inherit the typical limitations associated with GAN-based formulations.

The discussion of Integral Probability Metrics (IPMs) correctly notes their correspondence to different functional classes, but the experiments focus solely on Sup-MMD, essentially a standard MMD extended to a family of kernels. The practical comparisons between Sup-MMD and regular MMD flows [3] is not presented, which weakens the conceptual contribution.

The experimental setup appears to follow that of [4], but this work is not explicitly cited. Moreover, the paper reports only qualitative results (Figure 3) without quantitative metrics or comparisons to relevant baselines. Although the authors emphasize that their goal is not to outperform diffusion or conditional flow-matching models but rather to generalize transport-based sampling to broader objectives and implicit targets, including comparisons with related methods (e.g., MFLD or WFR) would significantly strengthen the empirical evaluation. The rationale for the chosen set of baselines also remains unclear.

---

### **Summary**
The paper is built on several established ideas and includes some interesting insights; however, the overall contribution is not clear and together with the experimental results seem to be not compelling enough to justify acceptance at the conference in their current form.

### **Misprints and Minor Issues**

- In the definition of the Wasserstein gradient, $\mathcal{F}$ is sometimes replaced by $\varphi$ (lines 091–097). It would be more consistent to use $\nabla_W \mathcal{F}$ throughout.

- Line 378: “subsubsection” should be replaced with “subsection.”

- Line 399: there is a duplicated word “Additionally.”

[1] Mokrov, Petr, et al. "Large-scale wasserstein gradient flows." Advances in Neural Information Processing Systems 34 (2021): 15243-15256.

[2] Alvarez-Melis, David, Yair Schiff, and Youssef Mroueh. "Optimizing functionals on the space of probabilities with input convex neural networks." arXiv preprint arXiv:2106.00774(2021).

[3] Arbel, Michael, et al. "Maximum mean discrepancy gradient flow." Advances in Neural Information Processing Systems 32 (2019).

[4] Korotin, Alexander, Nikita Gushchin, and Evgeny Burnaev. "Light Schrödinger Bridge." The Twelfth International Conference on Learning Representations.

**Questions:**

- In lines 205–207, the authors state that _“these approaches, however, primarily impose smoothness in the information-geometric sense.”_ It would be helpful if the authors could clarify what this statement means and elaborate on its implications in the revised version of the paper.

- Furthermore, the relationship between the proposed method and prior works [1], [2], and [3] should be explicitly discussed to better position the contribution within the existing literature.

- It would also strengthen the experimental evaluation if the authors reported quantitative metrics, such as accuracy scores comparable to those in [4], for the image generation task, and included comparisons with MFLD and WFR under the same experimental setup. The qualitative results in Figure 3 indicate limited diversity among the generated samples; the authors should clarify whether their method can control or improve sample diversity.

- Finally, it would be valuable to include additional instances of the proposed Integral Probability Metric (IPM), such as the Total Variation Distance, Wasserstein-1 distance, and standard MMD, in addition to the Sup-MMD case.

[1] Tankala, Chandan, Dheeraj M. Nagaraj, and Anant Raj. "Beyond propagation of chaos: A stochastic algorithm for mean field optimization." arXiv preprint arXiv:2503.13115(2025).

[2] Das, Aniket, and Dheeraj Nagaraj. "Provably fast finite particle variants of SVGD via virtual particle stochastic approximation." Advances in Neural Information Processing Systems 36 (2023): 49748-49760.

[3] Salim, Adil, and Peter Richtarik. "Primal dual interpretation of the proximal stochastic gradient Langevin algorithm." Advances in Neural Information Processing Systems 33 (2020): 3786-3796.

[4] Gazdieva, Milena, et al. "Light unbalanced optimal transport." Advances in Neural Information Processing Systems 37 (2024): 93907-93938.

---

### Official Review · Reviewer_Hgj2 · 2025-10-31

**Soundness:** 2
**Presentation:** 2
**Contribution:** 1
**Rating:** 2
**Confidence:** 4

**Summary:**

The authors study Stochastic Particle Gradient Descent (SPGD) and establish new convergence results under geodesic convexity and the Polyak–Lojasiewicz (PL) condition, incorporating explicit control over the bias and variance of the gradient estimator. They then introduce an additional regularization term to SPGD to address the vanishing-gradient problem that arises when source and target distributions have distant supports. The proposed regularization is evaluated on synthetic datasets and a male-to-female face translation task, demonstrating that objectives relying solely on the KL divergence fail to converge to the desired target distribution. Finally, the authors conduct multi-objective optimization experiments, showing that SPGD outperforms other particle-based methods.

**Strengths:**

An important aspect worth highlighting is that most of the theoretical statements are clearly explained and supported with intuitive discussions and examples.

**Weaknesses:**

**Clarity**

1. The authors frequently use the term “our” (e.g., “our formulation” in line 43 and “our proposed framework” in line 64, “our algorithm … our approach” in line 377); however, most of these references do not correspond to genuinely novel components. It seems that the only original contributions presented are the new convergence results and the introduction of the SPGD regularizer combined with the KL loss. This raises the question: what exactly is your proposed framework, and how is it defined?

2. No algorithms are presented in the main text. Only a special-case example (KL with Sub-MMD) in the appendix.

3. Since the proposed framework is unclear, it is difficult to understand the claim in line 65 that it “broadens the scope of transport-based sampling.” How exactly does it extend the existing SPGD framework?

3. Section 2 contains numerous definitions and mathematical constructions, but the presentation suffers because the authors list these objects without first explaining their purpose or how they relate to the paper’s main contributions. For example, in the SPGD subsection, it would be clearer to begin with the overall goal of the approach and then introduce necessary concepts such as differentiability in Wasserstein space. Similarly, the assumptions are presented before their meaning is explained. This narrative structure causes confusion, as it is initially unclear why these elements are introduced, and it limits the readability for a broader audience not deeply familiar with the underlying mathematics.

5. In Example 2, the authors do not explicitly state that the PL condition is obtained. It appears that the entropy regularization is meant to lead to this property, but the connection is not clearly articulated.

6. You state in line 50 that the studied framework can handle implicitly defined target distributions without requiring samples, but this point is not discussed or elaborated on later in the paper.

Given these points, it becomes clear that the authors do not effectively present their contribution. Even considering the positive aspects noted in the strengths, these issues highlight a lack of clarity in the paper.

**Theory**

1. The convergence analysis of SPGD is conducted in the infinite-particle setting, which, while preserving the theoretical significance of the results, nonetheless limits their direct applicability to practical, finite-particle algorithms.

2. In Theorem 1, the convex case $c = 0$ is examined only under the unbiased setting $\tau = 0$, which somewhat limits the generality and practical relevance of the result.

**Practice**

1. A central issue is the **weak link between theory and experiments**. The experiments apply SPGD to distant-support distributions, yet the theory suggests the KL loss should suffice. The authors then report practical failures of KL and add an Integral Probability Metric (IPM) term. In other words, the introduction of IPM is a solely practical need which has no relation with the previously introduced theory. Theory comes separately from the practice, theoretical aspects are supported/illustrated in the experiments.

2. Another issue with the enrichment of the loss for handling distributions with distant supports concerns its novelty. The idea of considering different loss functions $\mathcal{L}$ doesn’t seem as a new idea and has been explored in several prior works (see, for example, [1, 2]).

3. The estimation of the IPM gradient requires solving an additional optimization problem at each time step $t$, which limits the scalability of the method.

4. The multi-objective optimization (MOO) experiments add little novelty: the authors essentially run the known SPGD algorithm to approximate the Pareto front, comparing with previous methods on this task.

**Typos**

1. The heading “Subsection 4.0.1” appears to be misdefined; it should likely be “Subsection 4.1” rather than a latex subsubsection.

2. There are notation inconsistencies for the vector $x$: in Equations 16 and 17 it is written in regular font, whereas in line 390 it appears in bold $\mathbf{x}$.


[1] Arbel, Michael, et al. "Maximum mean discrepancy gradient flow." Advances in Neural Information Processing Systems 32 (2019).

[2] Aubin-Frankowski, Pierre-Cyril, Anna Korba, and Flavien Léger. "Mirror descent with relative smoothness in measure spaces, with application to Sinkhorn and EM."

**Questions:**

1. In lines 374–375, you state that you “assume” the MOO problem satisfies the PL condition. Why is this only an assumption rather than a property that can be shown to hold?

2. Are there any theoretical benefits to using the IPM that the authors extensively employ in the paper (from the theoretical perspectives of the Section 2)?

---

### Official Review · Reviewer_BHiw · 2025-11-01

**Soundness:** 3
**Presentation:** 3
**Contribution:** 2
**Rating:** 4
**Confidence:** 3

**Summary:**

The work is dedicated to transport-based sampling, where the target measure is defined via an optimization problem. The paper studies stochastic particle gradient descent (SPGD) for a functional $\mathcal{L}$ using a (biased) stochastic approximation of Wasserstein gradient. The authors consider an analog of a Lipschitz smoothness condition and perform convergence analysis of the method under the assumption that either geodesic convexity or Polyak-Łojasiewicz condition holds. The work features applications of the approach to multi-objective optimization and particle transport.

**Strengths:**

1. The paper is well-structured.
2. Applying SPGD to multi-objective optimization provides a clear and illustrative use case.
3. A strength of the paper is that the conclusion explicitly addresses the limitations of the proposed approach.

**Weaknesses:**

1. The paper's contribution is somewhat incremental, as the analysis mostly follows standard techniques from convex analysis.
2. Discussion of Sup-MMD is rather brief and could be made more clear, detailed and explicit.
3. The literature review could be more comprehensive. The paper should be more clearly positioned with respect to prior work, including a discussion of previous analyses of biased stochastic methods in optimal transport (if any).

**Questions:**

After Theorem 2, it is written that "we can deduce that the iteration complexity of SPGD to achieve $\epsilon$-accurate solution is $O\left(\max \lbrace\frac{L \sigma^2}{c^2 \epsilon}, \frac{L}{c}\rbrace \log \frac{1}{\epsilon}\right)$, provide $\tau=c \epsilon$." Where does $\frac{L \sigma^2}{c^2 \epsilon}$ come from?

---

### Meta-Review · Area_Chair_pc7L · 2025-12-25

**Summary:**

Reviewers raised a number of concerns:
- Contribution is incremental: the algorithm was already proposed in prior works, and the theory is a simple adaptation of well-known results from convex optimization. The theory does not attempt to handle the more realistic finite-particle setting. The contribution is also not well-positioned relative to the literature.
- The theory and experiments are disconnected.
- Experiments are light.

The main issue is the lack of novelty. The authors did not provide a rebuttal, so all concerns remain, and the initial scores were not very high. The natural conclusion is rejection.

**Reviewer Concerns:**

None of the reviewer concerns were addressed.

**Reviewer Scores:**

There was no rebuttal, so the scores would remain unchanged. Actually, Reviewer u6LU suggested to lower the score from 6 to 4, in light of the other reviews.

---

### Decision · Program_Chairs · 2026-01-26

Reject